# Formation of Lymphoma Hybrid Spheroids and Drug Testing in Real Time with the Use of Fluorescence Optical Tweezers

**DOI:** 10.3390/cells11132113

**Published:** 2022-07-05

**Authors:** Kamila Duś-Szachniewicz, Katarzyna Gdesz-Birula, Emilia Nowosielska, Piotr Ziółkowski, Sławomir Drobczyński

**Affiliations:** 1Department of Clinical and Experimental Pathology, Institute of General and Experimental Pathology, Wrocław Medical University, 50-368 Wrocław, Poland; katarzyna.gdesz-birula@student.umw.edu.pl (K.G.-B.); piotr.ziolkowski@umw.edu.pl (P.Z.); 2Department of Optics and Photonics, Faculty of Fundamental Problems of Technology, Wrocław University of Science and Technology, 50-370 Wrocław, Poland; emilia.nowosielska@student.pwr.edu.pl

**Keywords:** 3D lymphoma model, optical tweezers, single-cell manipulations, cell co-culture, cell adhesion, lymphoma–stromal cell crosstalk, AMD3100 (plerixafor), doxorubicin (DOX), ibrutinib (IBR)

## Abstract

Interactions between stromal and lymphoma cells in the bone marrow are closely related to drug resistance and therapy failure. Physiologically relevant pre-clinical three-dimensional (3D) models recapitulating lymphoma microenvironmental complexity do not currently exist. In this study, we proposed a scheme for optically controlled hybrid lymphoma spheroid formation with the use of optical tweezers (OT). Following the preparation of stromal spheroids using agarose hydrogel, two aggressive non-Hodgkin lymphoma B-cell lines, Ri-1 (DLBCL) and Raji (Burkitt lymphoma), were used to conduct multi-cellular spheroid formation driven by in-house-developed fluorescence optical tweezers. Importantly, the newly formed hybrid spheroid preserved the 3D architecture for the next 24 h. Our model was successfully used for the evaluation of the influence of the anticancer agents doxorubicin (DOX), ibrutinib (IBR), and AMD3100 (plerixafor) on the adhesive properties of lymphoma cells. Importantly, our study revealed that a co-treatment of DOX and IBR with AMD3100 affects the adhesion of B-NHL lymphoma cells.

## 1. Introduction

B-cell non-Hodgkin lymphomas (B-NHLs) are the most common lymphoproliferative malignancy in Western countries [1]. Staging bone marrow biopsies are positive for up to 25% of high-grade lymphomas [2,3]. They are mostly associated with a poor prognosis, partly due to the protective interactions of lymphoma cells with mesenchymal stromal cells (MSCs) [4,5,6]. Understanding the role of MSCs in the microenvironment of B-NHLs has been recognized as one of the greatest challenges in treatment failure in haemato-oncology. Furthermore, the complicated molecular biology of MSCs and their correspondingly complex dynamics remain primarily unknown. In our previous work, we studied via optical tweezers (OT) the formation of nascent adhesion between MSCs and B-NHL lymphomas using a broad range of lymphoma cell lines [7], as well as patient-derived primary cells [8]. We found that the adhesive properties are cell-line specific, and it is possible to distinguish lymphoma from normal B-cells based on the adhesion to the stromal cell. The above work was performed in a two-dimensional (2D) culture, whereas three-dimensional (3D) organization and cellular microenvironments emerged as critical determinants of tumor pathogenesis and drug resistance [9,10].

Tumor spheroids are the 3D aggregations of cancer cells alone (monospheroids) or a mixture of multiple cell types (hybrid spheroids) broadly used as in vitro models of tumor growth [11,12], differentiation [13], and drug resistance [14]. Given that tumors are composed of multiple cell types, 3D co-culturing increases the complexity of tumor models [15,16,17,18]. We recently established that lymphoma cells and MSCs, when co-cultured on agarose hydrogels, self-aggregate into stabile 3D spheroids [19]. Interestingly, we observed that stromal cells form the core of the spheroid and are evenly surrounded by layered lymphoma cells. In this study, we attempted to recreate a lymphoma hybrid model using optical tweezers driven by the arguments described in the following paper. Although several parameters, such as spheroid size and cell number, are easy to track, insight into the initial phases of spheroid formation remains unattainable using standard techniques. Furthermore, cell–cell interactions are essential in the initial phase of spheroid formation, allowing compact cell aggregation [20]. OT enables the study of the early stages of hybrid spheroid formation, which cannot be achieved with standard bulk techniques. Notably, the number of attached cells and the time of nascent adhesion formation may be strictly controlled. Moreover, with OT, we can precisely follow the minute changes in the adhesion caused by therapeutics [21]. Finally, gaining insight into the initial stage of growth will benefit the better understanding of the behavior of lymphomas at subsequent phases.

In this study, we attempted to form de novo a hybrid model of B-NHL and MSCs to study the initial stages of lymphoma–stromal cell adhesion. A single B-NHL lymphoma cell was trapped, optically seeded on a 3D stromal spheroid, and kept in direct contact until a stable connection was formed. The attachment time of a single cell was estimated as previously described [7,8,21]. The procedure was repeated until the entire surface of the stromal spheroid was covered with lymphoma cells. Next, we used fluorescence to assess the viability of the cells after the optical manipulations. Finally, and significantly, we examined the changes in lymphoma cell adhesion caused by anticancer drugs. We tested agents commonly used in B-NHL treatment chemotherapeutics, such as doxorubicin, as well as targeted drugs, such as ibrutinib and AMD3100. Importantly, it was shown that the AMD3100/DOX and AMD/IBR combinations led to higher synergy and decreased lymphoma cell adhesion to the stromal spheroid for both cell lines.

## 2. Materials and Methods

### 2.1. Optical Tweezers with Fluorescence Detection

Figure 1A shows a scheme of the in-house-developed fluorescence optical tweezers (FOT) based on the Olympus IX71 inverted biological microscope (Olympus, Hamburg, Germany). A strong optical trap is generated by a Nd:YAG 1064 nm laser (maximum output power 4W), and the Galvano-mirror XY scanning system enables control of the position of the trap in the sample. In the presented setup, we used two different kinds of cameras. Camera 1 (CAM1) is a fast and precise CMOS camera (MC1362, Mikrotron GmbH, Unterschleißheim, Germany) used in the calibration process [22] and in imaging samples during trap manipulation, as shown in Figure 1B. Second is the low-noise and high-sensitive camera 2 (CAM2, IRIS9, Photometrics, Tucson, AZ, USA), which is part of the fluorescence detection system, as shown in Figure 1C. Instead of the popular mercury arc lamps, in our setup, we used high-power LED with a peak wavelength of 490 nm. The set of fluorescence filters consists of excitation filter Exc.F (FESH0500, Thorlabs, Mölndal, Sweden), emission filter Em.F (FELH0550, Thorlabs), and dichroic mirror DM2 (MD499, Thorlabs). The FOT enables the forming of cellular spheroids and real-time fluorescence imaging.

### 2.2. Cell Lines and Culture Conditions

The human bone marrow cells HS-5 were obtained from American Type Culture Collection (ATCC, Rockville, MD, USA). The DLBCL cell line Ri-1 and BL cell line Raji were received from Deutsche Sammlung von Mikroorganismen und Zellkulturen (DSMZ, Braunschweig, Germany). All cell lines were grown in RPMI 1640 (with phenol red) medium supplemented with 10% fetal bovine serum (Gibco, Paisley, UK) and 1% penicillin/streptomycin (Gibco, Paisley, UK) in a humidified incubator at 37 °C and CO_2_. Cells were in the logarithmic growth phase at the beginning of all experiments.

### 2.3. Preparation of Gels and Mesenchymal Stromal Spheroids

The microwell hydrogels were prepared by incorporating 2% agarose (*w*/*v* in PBS, UltraPure™ Agarose, Life Technologies, UK) into microforms according to the manufacturer’s instructions (3D PetriDishR^®^, Microtissues Inc., Providence, RI, USA). Stromal cell spheroids were obtained by seeding 3.2 × 10^4^ of HS-5 cells in 190 μL of medium per mold (seeding densities of 125 cells/well). The stromal spheroids were grown 72 h prior to manipulations in optical tweezers in RPMI-1640 (Gibco, Paisley, UK) supplemented with 10% fetal bovine serum (FBS, Gibco, Paisley, UK) and antibiotics (Sigma-Aldrich Chemie GmbH) at 37 °C in a humid atmosphere saturated with 5% CO_2_.

### 2.4. Formation of Multicellular Hybrid Spheroids in Optical Tweezers

A three-day-old stromal spheroid was transferred from the agarose mold to an uncoated glass-bottom dish, and approximately 1 × 10^3^ of lymphoma cells in 100 μL of culturing medium were added. A single lymphoma cell was selected and optically trapped with the 100 mW power laser at the entrance of the microscope objective. Next, the microscope stage was moved to deliver the lymphoma cell into contact with the stromal spheroid surface. As soon as cell–spheroid contact was initiated, the B-cell was retained in the center of the optical trap to interact with the stromal spheroid until an adhesive junction was formed. The evidence of the formation of the adhesion junction between the lymphoma cell and the stromal spheroid was confirmed by three attempts to detach the lymphoma cell by the optical trap. If the cellular contact was broken, the lymphoma cell was again held in the optical trap to interact with the spheroid surface for a longer time. An individual cell was subjected to a maximum of three attachment attempts. The process was continued with the subsequent lymphoma cells until the entire surface of the stromal spheroid was covered by lymphoma cells.

The precise measurements of the adhesion force, as well as the influence of the laser beam used on living cell viability, have been described in our earlier papers [7,8]. All experiments were carried out at room temperature, i.e., 25 °C. The spheroids formed in the OT were then transferred to an incubator and cultured in a hanging drop in RPMI-1640 (Gibco, Paisley, UK) with 10% bovine serum (Gibco, Paisley, UK) and antibiotics (Sigma-Aldrich Chemie GmbH) under standard conditions. After one day, control photos were taken. Spheroid images were captured using homemade software written in C++.

### 2.5. Hybrid Spheroid Viability by Fluorescence Microscopy

To determine cell viability after manipulation in optical tweezers, hybrid spheroids were stained with calcein-AM (Thermo Fisher Scientific, Karlsruhe, Germany) and propidium iodide (PI, 2 μg/mL, Sigma). After 15 min of incubation in the dark, spheroid viability was evaluated using fluorescence microscopy. Staining was performed immediately after the completed manipulation and 24 h after re-incubation of the spheroid in the RPMI 1640 medium. The fluorescence was detected at excitation/emission wavelengths of 490/520 and 535/617 for calcein and PI, respectively, and separately analyzed by ImageJ software (National Institutes of Health, Bethesda, MD, USA). The percentage of living and dead cells in spheroids was calculated by the corrected total cell fluorescence (CTCF) intensity [23]. This method determines the CTCF by subtracting out background signal, which is useful for comparing the fluorescence intensity between cells.

### 2.6. Evaluation of the Adhesive Properties of Lymphoma Cells

The lymphoma cells of Ri-1 and Raji cell lines were attached to the surface of the stromal spheroid for a given time. The experimentally established contact time intervals for single-cell trapping were 10, 20, 30, 40, 50, and 60 s. Next, the optical trap was moved 20 μm away from the lymphoma cell for 10 s. The formation of the adhesion junction between the lymphoma cell and the stromal spheroid was evidenced by the fact that the B-cell could not be optically pulled away from the stromal spheroid during three detachment attempts. If the lymphoma cell was detached, it was again held in the optical trap to interact with the surface of the spheroid for a longer time. The B-cell was placed in contact with the spheroid a maximum of three times, and the entire time of an individual cell manipulation did not exceed 90 s. The adhesive properties of 30 cells from each experimental group (untreated and drug-treated cells) were evaluated in two independent experiments.

### 2.7. Drugs and Treatment

The optimal time and dose of ibrutinib (IBR) and doxorubicin (DOX) on Ri-1 and Raji cell lines have been previously investigated [19]. The ibrutinib IC_50_ for inhibition of Ri-1 and Raji cells proliferation was 0.514 μM and 6.18 μM, respectively, while the values of IC50 for DOX were 0.833 μM and 0.174 μM for Ri-1 and Raji cell lines, respectively.

The concentration of AMD3100 was chosen following Azab et al. [24]. DOX, IBR, and AMD3100 (plerixafor) were purchased from Sigma-Aldrich (Darmstadt, Germany). Stock concentrations for DOX (1 mM) and AMD3100 (5 mM) were made in water and stored at -20 °C, while IBR (10 mM) was dissolved in DMSO (Sigma Aldrich) and stored at 4 °C. Working stocks were made in culturing media. Ri-1 and Raji cells (0.5 × 10^6^) were grown in 1 mL of the medium supplemented with 1. AMD 3100 (50 μM), DOX (0.05 μg/mL), and IBR (0.4 μM) alone, or 2. DOX (0.05 μg/mL) with AMD-3100 (50 μM) and IBR (0.4 μM) with AMD3100 (50 μM) under standard conditions for 48 h. Appropriate untreated controls were prepared. Cells were washed in PBS to remove compounds, and cell viability was assessed by trypan blue staining in an automated cell counter and immediately analyzed with optical tweezers. Experiments were repeated twice for both cell lines and all of the tested drugs.

### 2.8. Statistical Tests

Statistical tests were performed using Microsoft Excel. Changes in the cell adhesion after the drug treatment were determined using Student’s t-test. *p* < 0.05 was considered to indicate a statistically significant difference, and values were expressed as the mean ± SD.

## 3. Results

### 3.1. The Recapitulation of the 3D Hybrid Spheroid in Real Time in Optical Tweezers

We successfully patterned the cellular architecture of a 3D hybrid spheroid in real time (Appendix A). Figure 2A–I shows the stages of the hybrid spheroid formation in optical tweezers. The stromal cell spheroid presented in Figure 2A was obtained from human immortalized HS-5 cells cultured on agarose microwells and served as a core of our 3D structure. In preliminary manipulations, we established that lymphoma cells of Ri-1 and Raji cell lines exhibited a high affinity to stromal cell spheroids, predominantly attaching in 10–20 s. Optical tweezers were used to capture the lymphoma cell (highlighted by the red box, Appendix A, Figure 2A) and moved toward the stromal spheroid via microscope stage motion. The cell was attached to the surface of the stromal spheroid for 10 s via active manipulation to allow for a proper membrane tether to form, as shown in Figure 2B. We initially positioned the single layer of lymphoma cells at the core of the stromal cell, as shown in Figure 2C–F. This structure was further expanded by attaching the second layer (Figure 2G,H), making a multilayered stabile 3D structure of approximately 65 cells, as presented in Figure 2I at the moment of 1:03:08.

Importantly, in order to make the three-dimensional structure of the spheroid, it was necessary to change the position of the optical trap in the volume of the microscopic sample. The position of the trap in the sample plane was controlled by the Galvano-mirror XY scanning system, while the change of depth was obtained by moving the microscope stage vertically. The distance of the optical trap from the microscope objective is constant and corresponds to the imaging plane. By changing the imaging plane using the micrometric microscope screw, we can change the position of the optical trap so that we can move the trapped objects more profoundly into the sample (Appendix A). 

After the manipulation was completed, the excess lymphoma cells not attached to the stromal mass were delicately pipetted, and the newly formed 3D structure was placed in a hanging drop in the RPMI medium and re-incubated under standard conditions. The resulting multicellular structure was highly stable for the first 24 h in the hanging drop culture, as evaluated by microscopy. However, in the following days of observation (2–7), the spheroid began to lose its original 3D shape and tended to flatten due to the increased proliferation of the lymphoma cells, which originally grew in suspension without forming tight connections between cells. In addition, the attempts to transfer the spheroid to the agarose gel with a pipette tip turned out to be ineffective and resulted in the dissociation of the lymphoma cells from the core of the stromal cells.

### 3.2. The Stability and the Viability of the De Novo Formed Spheroid

Here, we investigated if the optical trapping affected the viability of the newly formed spheroid. The cells were stained with calcein (live cells) and iodium propidine (PI, dead cells) immediately after the generation of the structure was completed and 24 h after the re-incubation. The visualization of the spheroids was achieved by adding fluorescent microscopy to our OT system. The representative images are shown in Figure 3A,B. We observed that the optical manipulation did not affect cell survival. Immediately after the end of the optical manipulation, the viability of the cells was 100% for both cell lines tested in our study. In addition, the pictures taken in fluorescence show the even surface coverage of the stromal spheroid with lymphoma cells. Significantly, we revealed that high cell viability was maintained the next day. The percentage of live cells 24 h after the re-incubation was 93% and 87% for Ri-1 and Raji cells, respectively, as shown in Figure 3C.

### 3.3. The Assessment of the Adhesive Properties of Aggressive Lymphoma Cell Lines

An important step in the metastatic spread of lymphoma is its adhesion to bone marrow stromal cells. We therefore first established the average time of nascent adhesion formation to a stromal spheroid. The contact time for adhesion to occur ranged from 10 to 20 s (mean of 11.7 ± 3.7) and from 10 to 30 s (mean of 15.2 ± 5.6) for Ri-1 and Raji cells, respectively (Figure 4A,B). A total of 83.3% of Ri-1 cells and 50% of Raji cells attached to the stromal spheroid in 10 s, which classifies both tested cell lines as highly adhesive. Interestingly, compared to our previous work [7], we observed that Ri-1 cells adhere faster to a 3D stromal spheroid, which is a more relevant in vitro model, than to the HS-5 cells growing as a monolayer.

### 3.4. The Influence of Combined Drug Treatment on Single-Cell Adhesion

In this work, based on precisely determined contact time values, we were able to observe the minute changes in adhesion resulting from the anticancer treatment. Ri-1 and Raji cells were incubated for 48 h either with single AMD3100, doxorubicin (DOX), or ibrutinib (IBR), or with a combination of AMD3100 with DOX and IBR. No significant effects of AMD3100, DOX, and IBR treatment on the viability of lymphoma cells were observed, except that IBR produced a significant reduction in Ri-1 cell viability. Moreover, no additive or synergistic effects were observed when combining AMD3100/DOX and AMD3100/IBR compared to the effects of the corresponding drugs alone; please see Appendix A. Comparable results with CXCR4 inhibitor have recently been described by Dragoj et al. on a non-small cell lung carcinoma model [25].

As assessed with optical tweezers, the treatment of Ri-1 and Raji cells with AMD3100 alone did not affect the adhesion of the lymphoma cells. Similarly, we did not observe any differences in the adhesive properties of cells incubated with DOX and IBR. Importantly, it was shown that the AMD3100/DOX and AMD/IBR combinations led to higher synergy and decreased lymphoma cell adhesion to the stromal spheroid for both cell lines. For Ri-1 cells treated with combined AMD3100 and DOX, the mean time required to establish adhesive interactions was 16.7 ± 6.99. It increased significantly when compared to DOX-treated cells (*p*-value < 0.01) and controls (*p*-value < 0.05), as shown in Figure 4A. The effect of the combined treatment was even higher for IBR. AMD3100/IBR treated cells adhered to the stromal cells spheroid 1.82 and 2.02 times slower than IBR-treated cells and controls, respectively. The adhesive properties of Raji cells were also significantly affected by the combined drug treatment, as presented in Figure 4B. The strongest increase in adhesion was observed after the AMD3100/DOX treatment as compared to Ri-1 cells. In turn, the incubation with AMD3100/IBR induced a similar range of changes in both of the cell lines.

## 4. Discussion

Optical tweezers are a micromanipulation tool that provide an unparalleled opportunity to trap, move, and connect living cells with focused laser light without direct physical contact. Broad applications of the tool include the study of the biophysical properties of trapped biomolecules [26], measurements of the stiffness of cancer cells and organoids [27,28], cell trapping inside living organisms [29], and studying the direct interactions between tumor cells and their microenvironment [30,31]. In the last decade, 2D and 3D cellular arrangement and structuring has garnered increasing interest [32,33]; however, the full potential of OT remains to be realized in this field. Moreover, optical tweezers have the capacity to position living cells accurately in three dimensions, as well as to determine the microscopic structure–function relationship between different cell types.

The first attempts to create a permanent 3D configuration of cells at predefined positions at the microscale were undertaken by Jordan P et al. in 2005 [34]. The authors manipulated individual E. coli within a gelatin matrix using holographic optical tweezers and confirmed the flexibility of this approach by arranging isolated cells in a variety of complex patterns. Ten years later, using mouse embryonic stem cells, 3D structures of varying geometries were created by holographic optical tweezer-based micromanipulation and stabilized by hydrogels [35]. Importantly, complex co-culture micro-environmental analogs were also generated to reproduce structures found within adult stem cell niches. In 2017, Yoshida A et al. optically assembled mouse mammary gland epithelial cells (NMuMG), as well as mouse brain neuroblastoma cells (Neuro2A), into various examples of 3D clusters in the presence of dextran [36]. The authors reported that 300 s of forced contact produced stable cell–cell contact. Importantly, they predicted the usefulness of the 3D assembly of neuroblastoma cells for modeling neuronal differentiation in 3D cell structures, as well as for demonstrating the relationship between the 3D cell positioning of undifferentiated neuronal stem cells and neurogenesis. All the above works tended to use additional matrices, such as hydrogel, gelatin, or dextran, to fix the position of individual cells and stabilize the 3D structure. However, the physicochemical properties of the above materials may affect the original adhesive properties of living cells [37]. In this work, we attempted to optically recapitulate the 3D hybrid lymphoma–stromal cell structure by attaching lymphoma cells to a previously prepared 3D stromal spheroid. Importantly, any other external elements were entered into the experimental setup to stabilize the patterned structure. These spheroids formed in real time reflected the self-aggregation of lymphoma and stromal cells in the hydrogel culture, as we recently described [19].

Lymphoma cells tend to spread to the bone marrow, potentially resulting in life-threatening complications. It is well established that direct lymphoma–MSC interactions are those essential for the chemoprotection of lymphoma cells by the bone marrow niche [38,39]. The HS-5 cell line used in our experiments has been recognized to reproduce the MSC capacity to influence tumor biology, as well as to evaluate the molecular mechanisms underlying the tumor immune escape mediated by stromal cells [40]. Having established the capability of OT to position a single lymphoma cell onto a monolayer of HS-5 stromal cells and study adhesive properties [7], we then moved on to testing the capacity of optical tweezers to physically form a multicellular spheroid in real-time mode. The positioning of multiple cells in 3D was accomplished by introducing several modifications to our original system. First, the optical tweezers setup was optimized to guide only one high-energy laser beam. Next, the optomechanical system was simplified, and the optical elements were selected to obtain high-efficiency infrared beam 1064 nm transmission. Finally, the Olympus UPlanSApo 60×/1.20 W (water immersion) microscope objective was used to generate the optical trap. Due to larger optical transmission, the traps generated by this objective are stronger compared to the previously used Olympus UplanFLN 100×/1.3. The water immersion objective has less chromatic aberration and allows the manipulation of the cells much farther away from the bottom surface of the glass bottom dish. In addition, lower magnification allows for observing a larger field in the sample, which is essential when working with large objects, such as spheroids. The use of LEDs as light sources for imaging and fluorescence stimulation creates an electronic way, without mechanical shutters, to conveniently switch between these two imaging modes. It is important during the experiment that the fluorescence signal is constantly being watched. The high-sensitivity and high-resolution camera allow for good-quality fluorescence observation, especially when we do not use laser sources for excitation and the fluorescence signal is low.

Originally, the manipulation of living cells in real time was significantly restricted due to the damaging effect of the lasers [41,42]. This limitation was overcome by the transition to infrared lasers, which enabled longer manipulation times of living cells, resulting in the broader use of optical tweezers in cell research [43]. In our previous works, we investigated the effect of a low-intensity optical laser on living lymphoma cells [7,8,22]. We proved that laser operation at 1064 nm poses a low risk of optical damage to lymphoma cells. Furthermore, we observed that lymphoma cells can be effectively trapped for over 10 min using a laser output power of 100 mW without showing any signs of cell damage. The results of fluorescent staining performed in this work confirmed no mechanical damage to the trapped cells at the point of contact.

Traditional ‘bulk’ adhesion studies on thousands of cells in a single experiment can only provide averaged results regarding cell behavior [44]. Precision in controlling the minute changes of single-cell adhesion in a time-dependent manner provides an invaluable OT tool. Since drug resistance is predominantly dependent on direct cell-to-cell contact, the second objective of the present study was to investigate the effect of AMD3100, DOX, and IBR treatments on the early adhesion of a single B-NHL cell to a stromal spheroid.

AMD3100 is one of the most common inhibitors of the chemokine receptor type 4 (CXCR4) [45]. Changes in CXCR4 expression have been linked to the progression and metastases of several cancers, including B-NHL [46,47]. CXCR4 allows migration and homing of the neoplastic cells to the regions where non-malignant stromal cells express the chemokine CXCL12 [45]. CXCL12 (also known as stromal cell-derived factor 1, SDF1), the ligand of CXCR4, is expressed in many human tissues, including the bone marrow. SDF1 is strongly chemotactic for lymphocytes and is associated with an unfavorable outcome in B-NHL [48]. AMD3100 can mobilize cancer cells away from their protective microenvironment, making them more accessible to conventional therapies [49]. While the effect of AMD3100 on cell mobilization from MSCs and migration is well documented, the results regarding its effect on cell adhesion remain conflicting. The findings of this work show that AMD3100 does not affect the ability of a single B-NHL cell to adhere to a stromal spheroid. This observation is in agreement with the results of a ‘bulk’ adhesion assay [50], as well as a single-cell approach previously performed on leukemia cells [51]. Significantly, Hou et al. proved a direct relationship between AMD3100 treatment and the decrease in the adhesion between leukemia and stromal cells using optical tweezers. The authors observed that AMD3100 indirectly weakens the adhesion between leukemia cells and adhesion molecules and significantly decreases the stiffness of leukemia cells. Discrepancies between our results and other studies, where the decrease of cellular adhesion upon the AMD3100 treatment has been noticed [24,52], may partially result from the use of ‘bulk’ adhesion methods and different tumor models.

Previously, it was found that combining AMD3100 with anti-cancer drugs showed enhanced efficacy in the treatment of various human cancers in preclinical and clinical studies by disrupting the protective interactions between malignant hematopoietic cells and the bone marrow microenvironment [53,54,55,56,57]. In this work, we demonstrated that the combined treatment of AMD3100 and DOX decreased the adhesion of B-NHL lymphoma cells to a stromal spheroid. With regard to our work, Regenbogen et al. established that the AMD3100/DOX combination reduced the adhesion of rhabdomyosarcoma cells far more strongly than either treatment alone [58]. Notably, doxorubicin significantly reduced the migration of RH30 cells, and this effect was amplified synergistically with the addition of AMD3100. A possible explanation for this observation may be that DOX increases CXCR4 expression in vivo and in vitro in different tumor models [59,60]. This is consistent with the inhibition of SDF1-driven migration induced by AMD3100/DOX reported in another study [61].

Similarly, it was revealed that AMD3100 combined with ibrutinib decreased the adhesion of lymphoma cells compared to treatment with a single IBR. The drug is the small-molecule inhibitor of Bruton’s tyrosine kinase (BTK), having a critical role in the oncogenic signal transduction pathway downstream of the B-cell receptor (BCR) in various B-NHLs [62]. It was shown that IBR disrupts cell adhesion between the tumor and its microenvironment in vivo through unknown molecular mechanisms [63]; however, our single-cell studies on lymphoma cell lines have not confirmed this observation. Knowing that the drug also inhibits the SDF1-induced cell adhesion to VCAM-1 [64], it is surprising that the effect of the combined AMD3100/IBR treatment on cell adhesion has not yet been studied so far. Our findings made with cell lines support the link between BCR signaling and cell adhesion. Notwithstanding, the phenomenon of the decreased adhesion of lymphoma cells after combined drug treatment deserves further investigation.

## 5. Conclusions

In conclusion, we showed for the first time that 3D lymphoma–stromal cell spheroids can be readily created using optical tweezers. Significantly, we were able to retain the 3D cultures for further biological analysis. The ability to generate a hybrid spheroid in real time provides novel insight into cells’ architectural complexity and microenvironments. Moreover, by mimicking in vivo interactions, it opens up a wide range of future applications, including personalized drug screening in lymphoma. Ideally, patient-specific cells should be used to study tumor–environment crosstalk. As our next step, we plan to demonstrate the preclinical utility of our in vitro bioengineered model using FNAB-derived primary cells for patient-targeted therapeutics screening.

## Figures and Tables

**Figure 1 cells-11-02113-f001:**
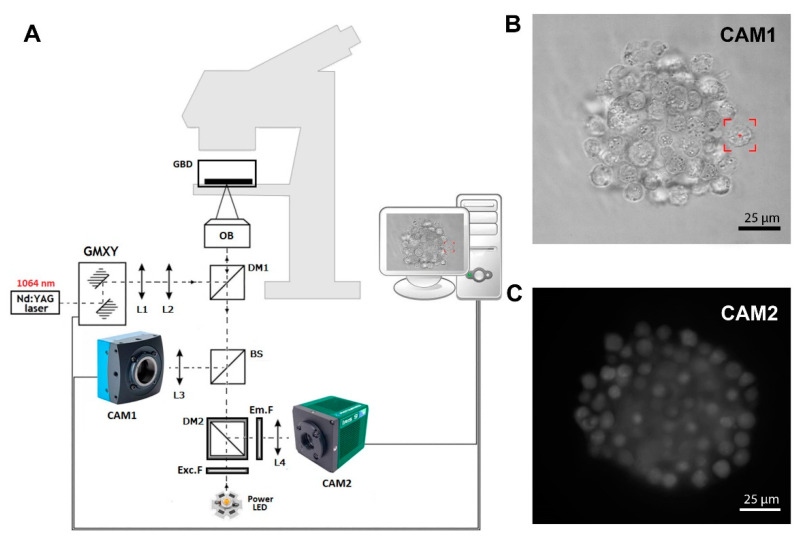
(**A**) In-house-developed fluorescence optical tweezers (FOT). L1, …, L4—lenses; DM1, DM2—dichroic mirrors; BS—beam splitter; GMXY—Galvano-mirror XY scanning system; OB—high NA numerical aperture objective; Exc.F—excitation filter; Em.F—emission filter; CAM1, CAM2—digital cameras 1 and 2. (**B**) The image of a hybrid spheroid taken in the brightfield by CAM1. The red box represents the optical trap. (**C**) The fluorescent image of a hybrid spheroid was taken by CAM2.

**Figure 2 cells-11-02113-f002:**
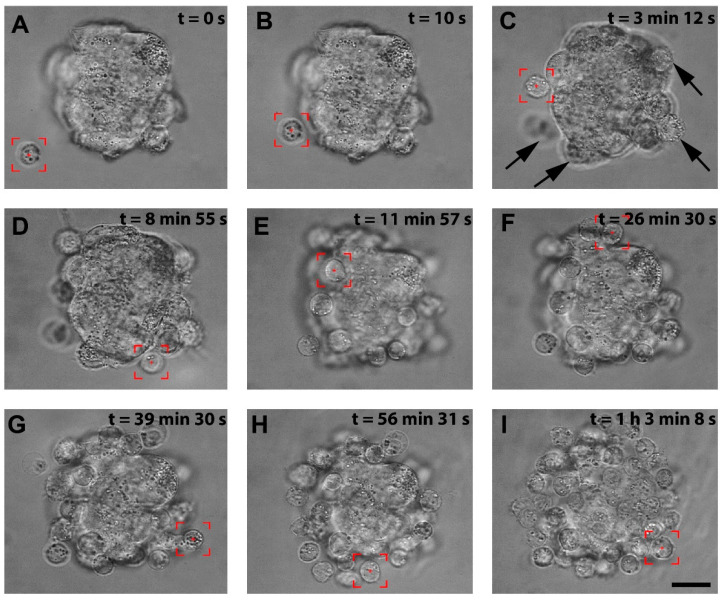
The visualization of the stages of the hybrid spheroid formation using OT. (**A**) A lymphoma cell is trapped by optical tweezers (highlighted by the red box). (**B**) the lymphoma cell is moved toward the surface of the stromal spheroid, and cell–spheroid contact is initiated. The lymphoma cell is retained in the center of the optical trap until an adhesion junction is formed between the lymphoma cell and the stromal spheroid. (**C**) Successive cells (indicated by arrows) are attached to the mass of the stromal spheroid. (**D**–**H**) The process of attaching subsequent lymphoma cells to the spheroid. (**I**) A complete hybrid lymphoma–stromal cell spheroid. Images were taken in the brightfield; scale bar = 25 μm.

**Figure 3 cells-11-02113-f003:**
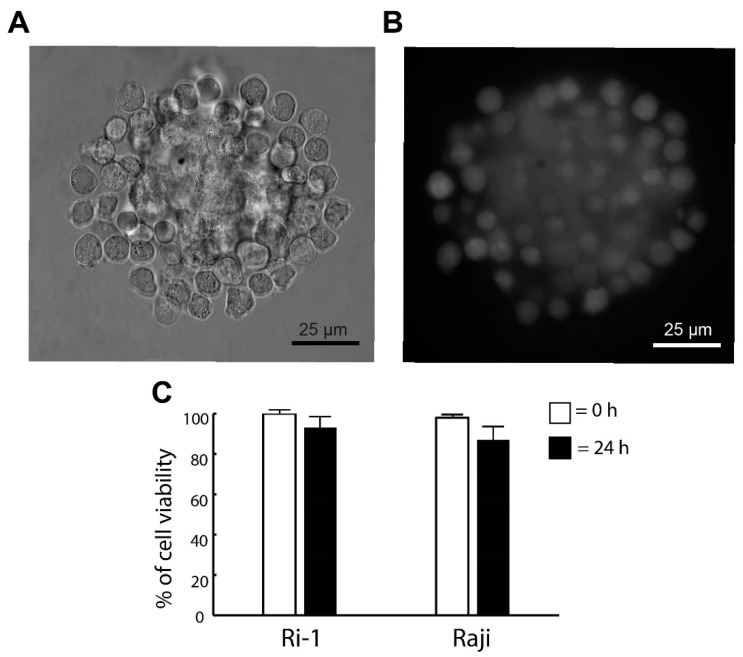
(**A**) De novo formed hybrid spheroid and (**B**) a fluorescent image of the spheroid stained with calcein (live cells) 24 h after the re-incubation. (**C**) Live/dead assay for the viability of the spheroid. Staining was performed immediately (0 h) and one day (24 h) after manipulations with calcein (live cells) and propidium iodide (dead cells). Quantification of the live/dead assay was performed using ImageJ software. An index of live lymphoma cells (% of cell viability) was constructed from the ratio of lived to the total cell numbers.

**Figure 4 cells-11-02113-f004:**
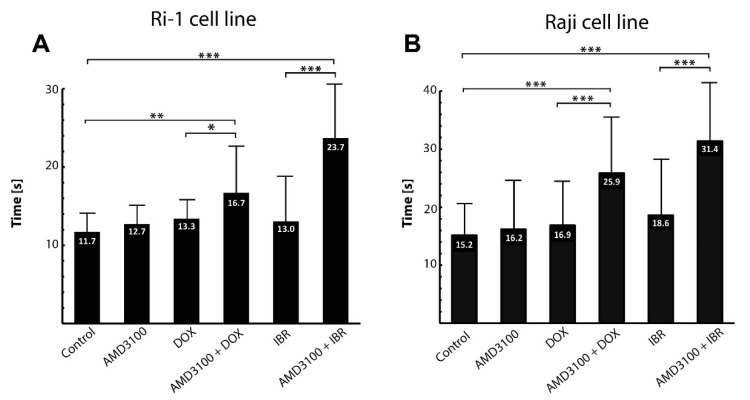
The effect of the anticancer treatment on the adhesion of Ri-1 (**A**) and RAJI (**B**) cell lines to the stromal spheroid evaluated in OT. Error bars represent the standard error of the mean calculated from three independent measurements. DOX, doxorubicin; IBR, ibrutinib. * *p*-value < 0.05, ** *p*-value < 0.01, *** *p*-value < 0.001.

## Data Availability

Not applicable.

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
