# Peer review of "Formation of Lymphoma Hybrid Spheroids and Drug Testing in Real Time with the Use of Fluorescence Optical Tweezers"

_cells, 2022, doi:10.3390/cells11132113_

Round 1
Reviewer 1 Report
The submitted manuscript presents an interesting study of the formation of hybrid lymphoma spheroids with particular interest on the adhesion forces between the lymphoma cells, manipulated by an optical tweezers, and the stromal spheroid. The authors validated the procedure observing the viability of lymphoma cells after the optical manipulation and eventually studied the effects of anticancer drugs on the adhesion forces.
The paper is well written and the results are convincing. I recommend it for publications.
I have only few minor comments.
The laser used is a 4 W laser. It seems to me an extremely high power. I don't think that 4 W is power at the sample. It has very well been established that at 1064 nm every 100 mW of power lead to an increase of 1 K at the sample.
This means an increase of about 40 degrees which is not realistic.
I suggest to measure the power after the objective lens to report a more realistic laser power.
Line 129 There is either an extra comma or a missing reference.
Lines 153-155. This is a curiosity driven comment. A lymphoma cell that doesn't adhere to the stromal spheroid within three attempts is discarded.
How? It is possible that the same cell will be captured eventually again?
Finally a comment of the discussion about the water immersion objective with respect to the oil one. (Line 325-329). The reason of the better performances is that water immersion objectives have less chromatic aberrations and allow to trap and to manipulate objects much farther away from the bottom surface of the sample cell. In addition the optical transmission is larger. Oil immersion objectives allow to manipulate objects within 20-30 microns from the surface.
Author Response
Dear Reviewer,
we would like to thank you for careful reading and discussing our article. We are very pleased with the positive impressions of our article. We especially appreciate your suggestion to explain the concerns regarding the power of laser used in our investigations. We answered all the comments addressed to our work. Please find enclosed a revision, which considers all comments.
- The laser used is a 4 W laser. It seems to me an extremely high power. I don't think that 4 W is power at the sample. It has very well been established that at 1064 nm every 100 mW of power lead to an increase of 1 K at the sample. This means an increase of about 40 degrees which is not realistic. I suggest to measure the power after the objective lens to report a more realistic laser power.
In the manuscript, we clarified that the power of 4W is the maximum laser output power, and in fact, the laser power during manipulation is less, as noted. The corrected laser power value has been updated. Please see lines 79-80, 118, and 353.
- Line 129 There is either an extra comma or a missing reference.
Thank you for your accuracy- an unnecessary comma was included in the text. The mistake have been corrected.
- Lines 153-155. This is a curiosity driven comment. A lymphoma cell that doesn't adhere to the stromal spheroid within three attempts is discarded. How? It is possible that the same cell will be captured eventually again?
Technically there was not possible to capture the same cell several times since cells that failed to attach to the stromal spheroid after three replications were transferred using an optical trap at a distance of at least 500 um from the manipulation zone and permanently attached to the glass bottom of the dish. Then we only selected free-floating cells for subsequent measurements.
- Finally a comment of the discussion about the water immersion objective with respect to the oil one. (Line 325-329). The reason of the better performances is that water immersion objectives have less chromatic aberrations and allow to trap and to manipulate objects much farther away from the bottom surface of the sample cell. In addition the optical transmission is larger. Oil immersion objectives allow to manipulate objects within 20-30 microns from the surface.
Following a suggestion, in the manuscript, a more precise explanation of the use of the 60x objective has been added. Please see lines: 333-337.
Reviewer 2 Report
Authors use optical tweezers to screen chemotherapeutic-anticancer drugs in real time. There are some concerns, that need to be answered:
According to authors,
“The optimal time and dose of ibrutinib (IBR) and doxorubicin (DOX) on Ri-1 and Raji cell lines were previously investigated [19]”
However, Ref 19 (DuÅ›-Szachniewicz K, Gdesz- Birula K, Rymkiewicz G. Development and Characterization of 3D Hybrid Spheroids for the 464 Investigation of the Crosstalk Between B-cell non-Hodgkin lymphomas and Mesenchymal Stromal Cells. OncoTargets Ther. 465 2022) is in press, thus the information is not accessible and data must be given.
The authors must also present data of the cytotoxic effect of these drugs, ibrutinib and doxorubicin, and the concentrations used, on the corresponding Ri-1 and Raji 159 cell lines (study of growth and death, at least for 24 hours), for comparison reasons. Furthermore, what is the IC50 of DOX and IBR for these cell lines?
There is also at least a wrong link.
According to authors, The percentage of living and dead cells in spheroids was calculated by the corrected total cell fluorescence (CTCF) intensity [23].
However, Ref 23 (University of Meryland, Baltimore Country. Available from: https://kpif.umbc.edu/image-processing-resources/imagej-fiji/de-termining-fluorescence-intensity-and-positive-signal/. Accessed May 8, 2022) must be checked, since it leads to a "PAGE can’t be find". Authors have to provide some details of the above methodology.
Author Response
Rewiever#2
Dear Reviewer,
we would like to thank you for taking the time to carefully reading and discussing our article. In particular, we appreciate the positive feedback from the Reviewer. We answered all the comments addressed to our work. Please find enclosed a revision, which considers the reviewers’ comments.
- According to authors, “The optimal time and dose of ibrutinib (IBR) and doxorubicin (DOX) on Ri-1 and Raji cell lines were previously investigated [19]” However, Ref 19(DuÅ›-Szachniewicz K, Gdesz- Birula K, Rymkiewicz G. Development and Characterization of 3D Hybrid Spheroids for the 464 Investigation of the Crosstalk Between B-cell non-Hodgkin lymphomas and Mesenchymal Stromal Cells. OncoTargets Ther. 465 2022) is in press, thus the information is not accessible and data must be given.
The article we are referring to in our present work just appeared in print; the reference has been updated; please see reference no. 19: DuÅ›-Szachniewicz K, Gdesz-Birula K, Rymkiewicz G. Development and Characterization of 3D Hybrid Spheroids for the Investigation of the Crosstalk Between B-Cell Non-Hodgkin Lymphomas and Mesenchymal Stromal Cells. Once Targets Ther. 2022;15:683-697. https://doi.org/10.2147/OTT.S363994.
- The authors must also present data of the cytotoxic effect of these drugs, ibrutinib and doxorubicin, and the concentrations used, on the corresponding Ri-1 and Raji cell lines (study of growth and death, at least for 24 hours), for comparison reasons. Furthermore, what is the IC50 of DOX and IBR for these cell lines?
In order to clarify the abovementioned issues, we prepared supplementary figure 1, where the viability data were presented, as well as we completed the Material and method section with the values of IC50 for DOX and IBR. Please see lines: 162-164.
- There is also at least a wrong link. According to authors, The percentage of living and dead cells in spheroids was calculated by the corrected total cell fluorescence (CTCF) intensity [23]. However, Ref 23(University of Meryland, Baltimore Country. Available from: https://kpif.umbc.edu/image-processing-resources/imagej-fiji/de-termining-fluorescence-intensity-and-positive-signal/. Accessed May 8, 2022) must be checked, since it leads to a "PAGE can’t be find".
An unnecessary dash appeared in reference 23, which made a redirection to the website impossible. The reference has been changed for: Determining Fluorescence Intensity and Signal. University of Meryland, Baltimore Country. Available from:https://kpif.umbc.edu/image-processing-resources/imagej-fiji/determining-fluorescence-intensity-and-positive-signal/. Accessed May 8, 2022
- Authors have to provide some details of the above methodology.
We completed the chapter 2.5. by the additional details regarding CTCF method, please see lines 145-146.
Round 2
Reviewer 2 Report
there is one more question to the authors
Since the DOX doses covered the IC50 values in the used cell lines (Ref 19), why in the Figure S1 we don't see this effect on growth?
Author Response
Dear Reviewer,
we would like to thank you for careful reading and discussing our article. We answered the additional comment addressed to our work. Please find enclosed a revision.
- Since the DOX doses covered the IC50 values in the used cell lines (Ref 19), why in the Figure S1 we don't see this effect on growth?
In the literature, doxorubicin concentration is usually expressed as µm/mL. In the present work, we intended to refer to the earlier research where DOX concentrations of 0.5 µg/mL and 0.05 µg/mL were used [1,2].
Indeed, a concentration of 0.05 ug/mL corresponds to 0.086 uM, which is approx. 10 and 2 times lower than IC50 values for Ri-1 and Raji cell lines, respectively. As we presented in Figure S1, this concentration does not affect the viability of lymphoma cells after 48h of incubation.
At the same time, I noticed that I did not mark statistical significance in graph S1 for the IBR treatment of Ri-1 cell lines. As in this case, the IC50 values were close to the drug dose, and we noticed a significant decrease in the viability of the Ri-1 line. Intriguingly, we did not see a statistically significant decrease in survival when IBR was combined with AMD3100, which we will try to explain in subsequent studies.
The changes have been put on the chart, and the text (lines 261-262) has been corrected. Sorry for the inconvenience.
- DuÅ›-Szachniewicz K, Gdesz-Birula K, Rymkiewicz G. Development and Characterization of 3D Hybrid Spheroids for the Investigation of the Crosstalk Between B-Cell Non-Hodgkin Lymphomas and Mesenchymal Stromal Cells. Onco Targets Ther. 2022 Jun 17;15:683-697.
2. An JH, Song WJ, Li Q, Bhang DH, Youn HY. 3D-culture models as drug-testing platforms in canine lymphoma and their cross talk with lymph node-derived stromal cells. J Vet Sci. 2021 May;22(3):e25